# Transgenerationally Transmitted DNA Demethylation of a Spontaneous Epialleles Using CRISPR/dCas9-TET1cd Targeted Epigenetic Editing in Arabidopsis

**DOI:** 10.3390/ijms231810492

**Published:** 2022-09-10

**Authors:** Min Wang, Li He, Bowei Chen, Yanwei Wang, Lishan Wang, Wei Zhou, Tianxu Zhang, Lesheng Cao, Peng Zhang, Linan Xie, Qingzhu Zhang

**Affiliations:** 1State Key Laboratory of Tree Genetics and Breeding, Northeast Forestry University, Harbin 150040, China; 2College of Life Sciences, Northeast Forestry University, Harbin 150040, China; 3Key Laboratory of Saline-Alkali Vegetative Ecology Restoration, Ministry of Education, College of Life Science, Northeast Forestry University, Harbin 150040, China; 4Shanghai Center for Plant Stress Biology, Center for Excellence in Molecular Plant Sciences, Chinese Academy of Sciences, Shanghai 201602, China

**Keywords:** NMR19, CRISPR/dCas9, DNA methylation, epigenetic inheritance

## Abstract

CRISPR/dCas9 is an important DNA modification tool in which a disarmed Cas9 protein with no nuclease activity is fused with a specific DNA modifying enzyme. A previous study reported that overexpression of the TET1 catalytic domain (TET1cd) reduces genome-wide methylation in Arabidopsis. A spontaneous naturally occurring methylation region (NMR19-4) was identified in the promoter region of the *PPH* (*Pheophytin Pheophorbide Hydrolase*) gene, which encodes an enzyme that can degrade chlorophyll and accelerate leaf senescence. The methylation status of NMR19-4 is associated with *PPH* expression and leaf senescence in Arabidopsis natural accessions. In this study, we show that the CRISPR/dCas9-TET1cd system can be used to target the methylation of hypermethylated NMR19-4 region to reduce the level of methylation, thereby increasing the expression of *PPH* and accelerating leaf senescence. Furthermore, hybridization between transgenic demethylated plants and hypermethylated ecotypes showed that the demethylation status of edited NMR19-4, along with the enhanced *PPH* expression and accelerated leaf senescence, showed Mendelian inheritance in F1 and F2 progeny, indicating that spontaneous epialleles are stably transmitted trans-generationally after demethylation editing. Our results provide a rational approach for future editing of spontaneously mutated epialleles and provide insights into the epigenetic mechanisms that control plant leaf senescence.

## 1. Introduction

Epigenetics is the study of heritable variations that occur in the absence of changes in the DNA sequence [1]. Epigenetic variations include DNA methylation, histone modification, and non-coding RNA. DNA methylation, the addition of a methyl group to position 5 of the pyrimidine ring of cytosine by the action of methyltransferase using S-adenosyl-l-methionine as the methyl donor, 5-methylcytosine (5mC) is inherited as cells divide [2,3], and is a biologically important epigenetic modification involved in gene regulation and transposon silencing [4,5,6], that plays crucial roles in genome stability, seed development, genetic imprinting, and abiotic stress in plants [7,8]. De novo DNA methylation is mediated by the RNA-directed DNA Methylation (RdDM) pathway in plants [9]. RdDM is initiated through the action of RNA POLYMERASE IV (Pol IV) to form single-stranded P4 RNA (Pol IV generates transcripts) [10,11,12], and then through RNA-DEPENDENT RNA POLYMERASE 2 (RDR2) to form double-stranded RNA (dsRNA) [13,14], which is cleaved by DICER-LIKE PROTEIN 3 (DCL3) to form 24 nt small interfering RNAs (siRNA) [15], and 24 nt siRNAs which are loaded into ARGONAUTE 4 (AGO4) [16,17,18]. The P5 RNA (Pol V generates transcripts) formed by RNA POLYMERASE V (Pol V) and 24 nt siRNA are complementary [19], recruiting DOMAINS REARRANGED METHYLTRANSFERASE 2 (DRM2) and initiating de novo methylation [20,21]. Unlike mammals, there are three contexts of 5mC DNA methylation in plants, respectively CG, CHG, and CHH (H represents A, T, or C), which are regulated by different methyltransferases [9]. After methylation is established, CG methylation is maintained by METHYLTRANSFERASE 1 (MET1), a homolog of DNA methyltransferase 1 (DNMT1) [9]. The methylation of CHG and CHH around the centromere and long transposon is maintained by CHROMOMETHYLASE 3 (CMT3) and CHROMOMETHYLASE 2 (CMT2) [22,23]. DRM2 is involved in the maintenance of CHH methylation at the edge of hetero-chromatin mid-long transposon and euchromatin mid-short transposon around centromere [22,23,24]. In plants, REPRESSOR OF SILENCING 1(ROS1) recognizes cytosine methylation and initiates DNA demethylation through the base excision repair process [25,26].

Previous approaches to the manipulate of DNA methylation and expression mainly relied on the mutations involved in the DNA methylation machineries or the chemical inhibitors of DNA methylation, leading to random genome-wide DNA methylation changes and the resulting massive off-target consequences [27,28,29]. Artificially mediated Zinc-finger has been used to fuse the RdDM effector such as Pol IV and RDR 2 on the *FWA* promotor [30]. The Clustered Regularly Interspaced Short Palindromic Repeats (CRISPR) system is widely used for targeted gene editing, and the inactivated endonuclease Cas9 has been used to achieve gene regulation and genome modification editing [31]. CRISPR/dCas9 is a system with transcriptional regulation function obtained by transforming the Cas9 protein in the CRISPR/Cas9 system into dCas9 (catalytically dead Cas9) [32]. Compared with the CRISPR/Cas9 system, the CRISPR/dCas9 system has the ability to bind to the target site without cutting the DNA strands [33], and the dCas9 protein and single guide RNA (sgRNA) are combined with some transcription factors to induce activation or inhibition of a given gene [34,35]. The demethylase Ten-Eleven Translocation (TET) enzymes from animals are involved in the demethylation process [36,37]. TET1 is an α-ketoglutarate and Fe^2+^-dependent dioxygenase that catalyzes the conversion of 5-methylcytosine (5mC) to 5-hydroxymethylcytosine (5hmC), and can also convert 5-methylcytosine (5mC) to 5-formylcytosine (5f C) and 5-carboxylcytosine (5ca C) [38], and the modified cytosine is specifically recognized and excised by thymus DNA glycosylase (TDG), and then is excised by the base excision repair pathway reverts to cytosine, which becomes the demethylation pathway [39]. Fusion of the TET1 catalytic domain (TET1cd) to the zinc-finger protein and dCas9 resulted in the loss of targeted gene methylation [40,41]. In plants, overexpression of TET1 cd in Arabidopsis leads to a decrease in CG methylation across the whole genome and can be inherited in subsequent generations [42]. Both the zinc-finger fusion to TET1cd and the dCas9-SunTag fusion to TET1cd with specific sgRNAs can specifically remove the methylation of the FWA gene promoter and FIE gene body, resulting in late flowering in Arabidopsis and a dwarf plant phenotype in rice, respectively [40,43]. Since CRISPR/dCas9 can be used for epi-editing, artificially altered DNA methylation modification in plants can be inherited and results in the formation of novel epialleles. Therefore, the introduction of the CRISPR/dCas9 system into plants is of great significance for the study of DNA methylation, regarding the causative connection between DNA methylation variants and observed phenotypic and/or expression variation in natural populations, namely spontaneous natural epialleles [44].

Differences in DNA methylation in specific regions of the genome cause changes in gene expression levels and phenotypes, and the alleles that the difference in the DNA methylation can be stably passed on to the subsequent generations are called epialleles [45]. The *SUP* (*SUPERMAN*) gene is a key gene that regulates flower development in Arabidopsis. Mutant *clk* (clark kent) and the *sup* mutant have similar phenotypes; that is, the flowers in both mutants have abnormal development. Experiments have shown that the *SUP* gene in the *clk* mutant has a lower expression level, but there is no difference in the nucleic acid sequence between the *clk* mutant and the wild type (WT). Bisulfite analysis showed that in the *clk* mutant, the SUP locus is highly methylated. Therefore, CLK and SUP are epialleles that can be stably passed on to subsequent generations [46]. Other studies on epialleles also include changes in fruit ripening, vitamin E accumulation, flowering time, photosynthesis, and sex determination [47,48,49,50,51]. Previously, we identified a naturally occurring methylated region (NMR19-4) [52], which is a LINE1 type transposon that is present in the promoter of some ectypes Arabidopsis *PPH* gene and is independent of the cis and trans genetic variations, so called pure epialleles [45,53]. In some ecotypes, NMR19-4 hypermethylation is associated with low *PPH* expression, while in other ecotypes, NMR19-4 hypomethylation is associated with high levels of *PPH* expression, resulting in different leaf senescence. Recently, a large number of DNA methylation variations have been identified between different natural accessions of Arabidopsis [54], and these be divided into two types: genetic variation-dependent methylation variation and genetic-independent methylation variation [55]. Our research focuses on the mechanism of methylation differences that are independent of genetic variation.

Overall, the epigenetic variation can originate from natural sources and induced approaches [56]. Natural sources of epigenetic variation refer to spontaneous epimutations, genetic changes (cis or trans), and Wide crosses polyploidy. Induced approaches of epigenetic variation include mutations in the epigenetic machineries (epiRILs), chemical treatments (5-AzaC, zebularine), epigenome editing, and tissue culture [56]. Thus far, methylation of *FWA* gene is regulated by other epigenetic factors, and the epi-edited plants are not subjected to Mendelian co-segregation analysis to test the trans-generational inheritance of edited methylation patterns. While the methylation of NMR19-4 is not regulated by the RdDM pathway and is independent of genetic variations, it provides a basis for us to study the heritability of epi-editing the spontaneous epialleles [52]. Here, we fused expression of a dCas9-TET1cd fusion protein in transgenic Arabidopsis plants from different ecotypes in which NMR19-4 is hypermethylated. We obtained plants with demethylated NMR19-4 in multiple subsequent generations, and the demethylated state can be in stable Mendelian inheritance without overall genome-wide changes in DNA methylation. Our results provide a reference for future study of trans-generational epi-editing spontaneous epialleles.

## 2. Results

### 2.1. Identification of NMR19-4-Related CRISPR/dCas9-TET1cd Epimutation

At present, studies of epigenetic editing in plants have mainly been focused on the FWA gene in Arabidopsis [30,40,57], which is regulated by the RdDM pathway, and little is known about the editing of naturally occurring epialleles. Previously, we identified a natural epialleles, NMR19, which is highly methylated in C24, but not in Col-0 [52]. The molecular mechanism of this spontaneous stochastic and genetic variation independent methylation formation is still unknown. In the TAIR10 reference genome, NMR19 is located at the 3′ end of the LINE1 transposon AT5G41835 on the chromosome 5. A specific truncated and inverted LINE1 transposon at 4.45 Mb on chromosome 5 of C24 was found by fine mapping and next-generation sequencing, which is the origin of NMR19-4. At the same time, 140 different ecotypes were screened, and the results showed that 20 ecotypes were hypermethylated in NMR19-4. The study found that NMR19-4 is located in the promoter region of the *PPH* gene, and different methylation states of NMR19-4 lead to differential expression of *PPH*; when NMR19-4 is hypermethylated, *PPH* expression is low, conversely, when NMR19-4 is hypomethylated, the expression of PPH is high. Therefore, we sought to further verify this conclusion by manually inducing demethylation, which showed that the expression of *PPH* is indeed affected by the methylation of NMR19-4. We randomly selected four ecotypes for demethylation editing among the 20 NMR19-4 hypermethylated ecotypes and screened the NMR19-4 methylation levels of these Arabidopsis ecotypes on the 1001 Genomes website (Figure 1a), and compared to NMR19-4 hypomethylated ecotype Ber, all screenshots of eco-types NMR19-4 that we selected for demethylation editing are hypermethylated. We designed three sgRNAs to target the two ends and the center of NMR19-4, and based on our previous information, the Pol III-dependent promoter At7SL-2 was used to drive the sgRNA expression [58], (Appendix A). Subsequently, the three different sgRNAs were cloned into the dCas9-TET1cd expression vector, respectively. Then, the CRISPR/dCas9-TET1cd vector was then used for plant transformation (Figure 1b). To determine whether the transformation caused methylation changes, we carried out Chop-PCR (Methylation Sensitive Restriction Enzymes PCR) with MspI methylation-sensitive enzyme based on our previous results [52], and TET1cd genotyping at the same time (Figure 1c). The results of Chop-PCR showed that only the DNA fragment from ecotype Krot 0 was cut, but not the other ecotypes, indicating that the epigenetic editing was successful (Figure 1c). Thus, we obtained Krot 0-65dc demethylated plant using the CRISPR/dCas9-TET1cd, indicating that our experimental system can be used for epi-editing.

### 2.2. Inheritance of the Epimutation in Krot 0-65dc

Whether the epi-edited alleles can be stably inherited through multiple generations is an important concern for epigenetic editing. To investigate the inheritance in Krot 0-65dc, we randomly selected 18 T2 generation individuals and 22 T3 generation individuals for Chop-PCR assay of the NMR19-4 methylation levels and found that in both the T2 and T3 generations, the progenies of the edited individuals remained de-methylated compared to the wild type (Figure 2a). We also screened homozygous edited individuals without TET1cd in the T4 generation, and we found that demethylation was still present in the NMR19-4 region in the edited individuals without TET1cd vector, demonstrating that demethylation editing can be inherited in the offspring independent of the vector (Figure 2b). Based on our previous results [52], the NMR19-4 transposon is located in the promoter region of the *PPH* gene and potentially affects the expression of the *PPH* gene through DNA methylation. To determine whether epi-editing-induced demethylation can lead to changes in *PPH* gene expression, we assayed the expression levels of *PPH* gene in the epi-edited individuals, the Krot 0 wild-type, and the other two NMR19-4 hypermethylated ecotypes C24 and Per 1. The results showed that the gene expression of *PPH* in the epi-edited plants from T2 and T4 generations were significantly higher than that of the Krot 0 wild type, C24, and Per 1 (*p* < 0.05, *t*-test, Figure 2c), demonstrating that epigenetically induced methylation changes affect the expression of the *PPH* gene. Therefore, the hypomethylation status of the methylation-edited NMR19-4 can be stably inherited in the progeny plants. Our results suggest that in the spontaneous epialleles, artificial epigenetic editing can affect the methylation changes and then regulate the expression of the related genes.

DNA methylation is the most stable and most extensively studied epigenetic modification and is mainly found to silence transposons, thereby inhibiting the expression of proximal genes. Since the methylation level of NMR19-4 is negatively correlated with the leaf senescence [52], we treated Krot 0-65dc T4 generation, Krot 0 wild type, Per 1, and C24 to dark induction, and recorded their leaf senescence phenotypes before and after 6 days of dark induction. The results showed that demethylation editing led to a decrease in the methylation level of the Krot 0-65dc T2, T3, and T4 generations (Figure 2a,b), which also led to the accelerated leaf senescence compared with the NMR19-4 hypermethylated ecotypes (Per 1 and C24), in which demethylated Krot 0-65dc T4 individuals exhibited accelerated dark-induced senescence and yellowing of leaves (Figure 3a). To confirm the accelerated leaf senescence induced by demethylation editing, we measured the chlorophyll levels; we found that the chlorophyll content of all plants was consistent before dark induction, while after dark induction, demethylation-edited individuals showed significantly lower chlorophyll content than others, strongly supporting the leaf senescence phenotype (*p* < 0.01, *t*-test, Figure 3b). This result is also consistent with our previously published results showing that hypomethylated NMR19-4 is associated with elevated *PPH* gene expression and accelerated leaf senescence [52]. We also performed bisulfite sequencing analysis of the NMR19-4 region; the result demonstrated that the target region showed obvious hypomethylation in all three contexts in T4 generation (Figure 3c, Appendix A). To further validate the targeted DNA demethylation of NMR19-4, we performed whole-genome bisulfite sequencing (BS-seq) of the two T4 generation individuals without TET1cd and two Krot 0 wild-type plants. Over 95% CT conversion in all of the samples by calculating the bisulfite mapping rate of unmethylated chloroplast and mitochondrial DNA, indicating that the bisulfite treatment was complete, and the results are reliable (Appendix A). In total, 42,115,420–52,859,086 raw reads were generated from each sample and the mapping rates over 75% for all the samples, providing high-quality methylation level data (Appendix A). Analysis of our sequencing results showed that our epi-editing did not cause a large change in the whole-genome methylation level (Figure 3d and Appendix A). At the same time, we showed that the methylation levels of C, CG, CHG, and CHH of NMR19-4 in T4 generation progeny plants was lower than that in the Krot 0 wild type (Figure 3e), corroborating our Chop-PCR results (Figure 1c and Figure 2a,b). We also sequenced the transcriptomes of T4 generation TET1cd-free individuals in the and Krot 0 WT, and found that *PPH* expression was significantly increased in the epi-edited progeny plants (Figure 3f, Appendix A). Thus, using our CRISPR/dCas9-TET1cd system, we demonstrated that targeted demethylation editing of NMR19-4 charged the demethylation status of offspring, and that is stably inherited and accelerates leaf senescence. Through methylation detection, quantification of *PPH* expression measurement, and analysis of leaf senescence phenotype analysis in the demethylation edited individuals, we also proved that *PPH* expression is indeed controlled by, but not just associated with, the methylation status of NMR19-4, illustrating the feasibility of our demethylation editing system.

### 2.3. NMR19-4 Methylation Status after CRISPR/dCas9 Editing Is Mendelian Inheritance

Since NMR19-4 is a spontaneously formed epiallele and is not regulated by RdDM [52], we wanted to ask whether the hypomethylation state is Mendelian transmission of alleles in crosses between the epi-edited Krot 0 hypomethylated individuals wild-type plants. To test our hypothesis, according to documents of Arabidopsis thaliana SNPs of different ecotypes and the methylation levels of NMR19-4, we selected F1 and F2 plants from reciprocal crosses in Krot 0-65-10-9-1dc (NMR19-4 hypomethylated individuals), Per 1 (NMR19-4 hypermethylated ecotype), and C24 (NMR19-4 hypermethylated ecotype) to examine whether both *PPH* expression and leaf senescence co-segregate with the NMR19-4 methylation status using Cleaved Amplified Polymorphic Sequences (CAPS), Chop-PCR, leaf senescence induced in dark, and qPCR assays (Figure 4b,c and Appendix A). Here, the methylation levels in F1 and F2 progenies from crosses between hypermethylated ecotypes and NMR19-4 hypomethylated individuals are specially inherited by allele, that is, there is no interaction between the methylation levels of different alleles, which is consistent with Mendelian segregation (Figure 4a,b). As a result, all F1 progenies were in a hypermethylated state, with the same level of *PPH* expression and chlorophyll content as in the parents (Figure 4b,c). In addition, among the F2 progeny, there were individuals that were homozygous for the NMR19-4 hypermethylated allele and individuals that were homozygous for the NMR19-4 hypomethylated allele. CAPS genotyping proved which parents NMR19-4 hypomethylated and hypermethylated alleles came from. Thus, the methylation status of NMR19-4 in demethylation-edited Arabidopsis plants is stably inherited (Figure 4b). For further validation, we examined chlorophyll content and *PPH* expression levels in some NMR19-4 hypermethylated and hypomethylated individuals from the F2 generation. Consistently, in the F1 hybrid and the segregating F2 population, the expression levels of *PPH* in plants in which NMR19-4 was hypomethylated were higher than those carrying the NMR19-4 hypermethylated allele (*p* < 0.05, *t*-test, Figure 4c), while the chlorophyll levels were lower, which was further validated by selecting different parents to cross (Appendix A). These results indicate that the methylation status of the epi-edited NMR19-4 can be stably separated in the hybrid progeny, and the edited NMR19-4 can also regulate the expression of *PPH* as well as leaf senescence in Arabidopsis. These results are consistent with those from our previous crosses between NMR19-4 hypermethylated and hypomethylated ecotypes, in which the F2 individuals from hypomethylated plants have low methylation levels, high levels of *PPH* expression, and rapid chlorophyll degradation, and vice versa [52]. Thus, we confirmed a Mendelian inheritance of epi-edited spontaneous epialleles by showing that the hypomethylated state of NMR19-4 can still be segregated in F2 progeny after de-methylation-edited plants are crossed with hypermethylated plants, and the hypomethylated state of NMR19-4 was not affected by hypermethylation counterparts via RdDM dependent trans-chromosomal methylation interaction in the F1 hybrids [59,60]. Taken together, our experimental results demonstrate that the methylation state of spontaneous epialleles produced by CRISPR/dCas9 editing can be stably maintained by trans-generational inheritance and further showed that methylation of NMR19-4 regulates leaf senescence in Arabidopsis.

## 3. Discussion

The CRISPR/Cas9 system consists of a single guide RNA (sgRNA) and the Cas9 protein and is an important tool for targeted modification of DNA [61]. In recent years, the CRISPR/Cas9 system has been modified to regulate transcriptional activity by fusing other enzymes. Cas9 is a protein composed of a NUC nuclease domain and a REC domain [62]. The NUC domain includes RuvC domain, PAM domain, HNH domain, and WED domain [62,63]. Cas9 protein uses the HNH domain and RuvC domain nuclease activity to perform cleavage function [61]. Firstly, the sgRNA recognizes the complementary sequence in the genome and form an RNA-DNA heteroduplex. Then, Cas9 introduces a double strand break (DSB), leading to that, the damaged DNA is repaired in the organism to generate mutants in the target sequence [64]. The CRISPR/dCas9 system is an upgraded system based on the CRISPR/Cas9 system that can be used to regulate and for targeted epigenetic editing. The HNH and RuvC domains of the Cas9 protein have been mutated to make inactivate [32]. The dCas9 protein without its original cleavage function can be fused to a variety of proteins, such as transcriptional activators or repressors and epigenetic enzymes, and is guided by sgRNA to modify the target DNA strand precisely [31]. Studies have shown that the expression of target genes can be efficiently induced or silenced by the fusion expression of dCas9 to the transcriptional activator VP64 or the transcriptional inhibitor KRAB [65,66].

Epigenetics includes histone modification, DNA methylation, and non-coding RNA regulation. Therefore, researchers have made a lot of attempts in epigenetic editing using CRISPR/dCas9 technology. In 2015, it was found that fusing the human acetyltransferase p300 with the dCas9 protein can promote the acetylation of the promoter H3K27, resulting in the activation of target gene transcription [67]. In 2017, dCas9-HDAC1 was used to modify histone deacetylation on the KRAS promoter, resulting in KRAS gene silencing [68]. Chen et al., fused histone methylase with dCas9 to form a CRISPR/dCas9-EZH2 system that can target histone methylation and downregulate gene expression [69]. The dCas9 SunTag-JARID1A histone demethylation system affects gene expression by reducing histone methylation levels [70]. The study found that dCas9 fusion to DNMT3a can induce specific promoter methylation and can also use multiple gRNAs to target multiple sites in order to methylate large fragments of promoter regions [71]. In plants, dCas9 fused to NtDRMcd targeted mutant FWA methylation and produced an early-flowering phenotype [65].

DNA demethylases play an important role in the regulation of epigenetic modification in organisms [72]. The study achieved targeted demethylation using the CRISPR/dCas9 system and successfully induced FWA gene activation [65]. Through genetic engineering, the MS2 RNA element was inserted into the sgRNA and cooperated with dCas9-TET11cd to successfully recruit more demethylases to achieve demethylation [73]. The modified dCas9-SunTag system successfully recruited more demethylases and achieved a wider range of demethylation. In plants, targeted demethylation of the FWA promoter using SunTag-TET1cd resulted in site demethylation and yielded a late flowering phenotype [40]. In our experiments, we attempted to demethylate NMR19-4 via editing with dCas9-TET1cd fusion, and further verified that *PPH* expression is regulated by NMR19-4 methylation. Our epi-editing experiments demonstrated that the methylation status of NMR19-4 affects the expression of *PPH*, which in turn affects the rate at which leaf senescence occurs. Therefore, based on this previous foundation and the results of our research, we can try to fuse the plant native demethylase, like ROS1 or DEMETER [72], into the CRISPR/dCas9 epi-editing system in the future.

The growth of plants is easily affected by their environment, so a series of self-protection mechanisms have evolved to maintain growth and development. Plants have evolved complex mechanisms that allow them to sense and adapt to changes in the external environment. Among these, DNA methylation plays an important role in the response of plants to environmental changes [54]. Drought induces altered methylation sites in rice, and some of these altered methylation sites were retained after stress relief [74]. The study found that when Brassica napus L. was treated with potassium dichromate, it caused genome-wide hypermethylation [75]. On the contrary, the whole genome methylation in Trifolium repens and Cannabis sativa decreased under stress such as Ni^2+^, Cd^2+^, and Cr^6+^ [76]. The methylation level in the potato cultivar ‘Russet Burbank’ was reduced after salt stress treatment, while DNA methylation was significantly increased in alfalfa in response to high-salt treatment [77,78]. Temperature is an important factor that impacts plant growth and development, and high temperature affects DNA methylation status in Arabidopsis [79]. Cold stress-induced hypomethylation of the maize root-specific Ac/Ds transposon region due to down-regulation of MET1 expression [80]. In future research, we can learn from the use of CRISPR/dCas9 epi-editing system to improve plant traits and make their growth more adaptable to environmental changes.

Leaf is an important photosynthetic organ in plants. SUVH2-overexpressing plants show abnormal leaf development and delayed senescence phenotypes [81]. Transcriptome analysis found that SUVH2 overexpression alters the inducible expression of genes that regulate aging [82]. In 2018, we discovered the transposon NMR19-4 by screening 140 ecotypes of Arabidopsis and that methylation of NMR19-4 can inhibit the expression of leaf senescence gene *PPH* and thus affect the leaf senescence in different ecotypes of Arabidopsis [52]. At the same time, the methylation level of NMR19-4 was significantly negatively correlated with the average temperature in the dry season. Therefore, NMR19-4 may adapt to environmental changes by changing the expression of *PPH* gene. Because leaf senescence is a complex biological process, the epigenetic regulation mechanisms are still unclear. For the current plant epigenetic editing, most of them involve in editing the commonly used FWA gene, which is regulated by RdDM [30,40,65]. Therefore, we selected NMR19-4 for editing because the spontaneous mutation is not regulated by RdDM. After the epi-edited plants were obtained, crosses were carried out to verify that the edited demethylation state could be transmitted to future generations in Mendelian inheritance. In-depth elaboration and study of spontaneous methylation variation plays an important role in increasing our understanding of epigenetic diversity. Research has emphasized the role of methylation variation in the response and plant stress adaptation [83].

Intriguingly, in the four NMR19-4 hypermethylated ecotypes that were randomly selected for CRISPR/Cas9-TET1cd targeted epi-editing, only Krot 0 was demethylated successfully. We also give the positions of the sgRNAs and PAM sequences in the four ecotypes (Appendix A), but we failed to find SNP differences in the positions corresponding to the sgRNAs of the four ecotypes. Since dCas9 SunTag-fused VP64 can activate transposon expression in heterochromatin regions [65], an ATAC-seq (assay for transposase-accessible chromatin using sequencing) analysis showed that whether NMR19-4 was hypermethylated or hypomethylated, its chromatin position was not open (Appendix A). However, why Krot 0 was edited successfully is still worthy of additional research. In future research, we will improve the vector to use a SunTag fusion vector, and try to fuse methylating enzymes such as DRM with dCas9 for epi-editing. We will also try to use different editing methods for the editing in plants to provide a basis for the subsequent improvement of crop traits. At the same time, demethylation of NMR19-4 by epi-editing has deepened our understanding of leaf senescence mechanisms.

With the in-depth application of CRISPR/dCas9, it has also made progress in plant research. At present, CRISPR/dCas9 is mainly used for gene expression regulation and epigenetic modification. For example, activation of FWA gene expression with dCas9 and VP64 in Arabidopsis [65], linking dCas9 to TET1 in Arabidopsis and rice promotes the reduction of target site methylation level [40,43]. Therefore, the CRISPR/dCas9 system has shown strong adaptability in plants. In our study, the CRISPR/dCas9-TET1cd demethylation system can target spontaneous epialleles-NMR19-4, leading to methylation reduction and can be stably inherited in the progeny. It provides a reference for future research on spontaneous epialleles in more crops, and also broadens ideas for future crop breeding.

## 4. Materials and Methods

### 4.1. Plant Materials and Growth Conditions

All plants were grown under 16 h light/8 h dark. Arabidopsis seeds were grown on 1/2 Murashige and Skoog (MS) medium solidified with 0.7% agar and containing 1.5% sucrose, and plates were incubated at 4 °C for 7 days in darkness before being transferred into a growth chamber. After 14 days of growth in the growth chamber, the 14-day-old seedlings were transferred to the soil. All mutant lines in this study were Krot 0 background. When Arabidopsis grows to flowering, the flowers were emasculated in the afternoon and pollination was performed in the morning of the next day, thereby producing the F1 generation hybrid seeds.

### 4.2. Vector Construction and PCR Assay

We used the Arabidopsis Pol III-dependent promoter At7SL-2 to initiate sgRNA transcription. TET1cd was cloned and linked into the 1300-dCas9 vector [41]. Three sgRNAs were linked into the 1300-dCas9-TET1cd vector, respectively. The final 1300-sgRNA-dCas9-TET1cd vector was transformed into GV1301 (*A. tumefaciens*), which was then transformed into the Arabidopsis plants. The seeds of T0 transgenic plants were screened on 1/2 MS with hygromycin B. We designed special primers to confirm TET1cd insertion, for Chop-PCR and identification of hybrid generations, and then MspI(NEB) for Chop-PCR and MboI(NEB) and EarI(NEB) for CAPS detection. Genomic DNA samples (100 ng) were digested for 4 h at 37 °C in a 20 μL reaction mixture volumes. For Chop-PCR, PCR detection was performed using 4 μL of the digest genomic DNA as template. All of our PCR reactions were carried out using 2 × Es Taq Master Mix (CWBIO, Taizhou, China). Primer sequences were listed in (Appendix A).

### 4.3. Whole-Genome Bisulfite Sequencing

For the preparation of Whole-Genome Bisulfite Sequencing (BS-seq) libraries, genomic DNA was extracted from two-week-old plants using CTAB-based method. After the Illumina second-generation sequencing was off, the raw data of DNA bisulfite sequencing was obtained. First, quality control of the raw data was performed using software SeqPrep (https://github.com/jstjohn/SeqPrep, accessed on 12 November 2021) and Sickle (https://github.com/najoshi/sickle, accessed on 12 November 2021). The clean reads after quality control were aligned to the Arabidopsis genome through Bsmap (https://github.com/genome-vendor/bsmap, accessed on 12 November 2021), and the methylation alignment rate and coverage were calculated. The false discovery rate (FDR) was calculated using the Bonferroni method to correct for *p* values, and FDR < 0.05 was to be considered statistically significant. Methylation levels of C, CG, CHG, and CHH contexts were calculated by using perl scripts.

### 4.4. Bisulfite Sequencing

Krot 0 WT and T4 generation genomic DNA (1 μg) were performed for the CT conversion using the EZ DNA Methylation-GoldTM Kit (ZYMO, Irvine, CA, USA). Then the CT conversed DNA template was for PCR amplify by using 2 × Es Taq Master Mix (CWBIO). The purified products cloned into pCE2 TA/Blunt-Zero Vector (Vazyme, Nanjing, China) and transformed into *Escherichia coli* DH5α. Total 20 positive clones were analyzed by using web-based Kismeth (http://katahdin.mssm.edu/kismeth, accessed on 15 May 2022) [84].

### 4.5. qRT-PCR Analysis

For all of our qRT-PCR experiments, total RNA was extracted from the leaf tissue of plants that had been grown in soil for 10 days. An amount of 1μg of total RNA was used for cDNA synthesis by cDNA synthesis kit (TaKaRa, Beijing, China). The cDNA was then diluted and used for qPCR validation of gene expression and then stored at −40 °C. We used the SYBR Green I Master mixture (Roche, Basel, Switzerland) as qRT-PCR reagent. The 2−ΔΔCT method was used to calculate the relative gene expression levels [85]. Primer sequences were given in (Appendix A).

### 4.6. RNA-Seq Analysis

Total RNA was extracted from 2-wk-old plants using based-TRIzol extracyion method. The original Illumina transcriptome sequencing data was checked, and the fastp software (https://github.com/OpenGene/fastp, accessed on 27 January 2022) was used for quality control to obtain high-quality clean reads. Sequencing reads were aligned to the TAIR10 using TopHat (http://tophat.cbcb.umd.edu/, accessed on 27 January 2022). After sequence alignment, the gene expression of all samples were calculated using the RPKM method.

### 4.7. ATAC-Seq Analysis

For the preparation of ATAC-seq (assay for transposase-accessible chromatin using sequencing) analysis, the quality control after sequencing was checked by FastQC (http://www.bioinformatics.babraham.ac.uk/projects/fastqc/, accessed on 21 April 2022). For data showing residual joints during quality control, we used fastp software to remove connectors [65]. The data were aligned to the *Arabidopsis thaliana* reference genome incorporating the NMR19-4 sequence using Bowtie2 software [86], and finally further converted into bam files. After the bam files were deduplicated with sambamba software [87], they were converted into wig files and visualized by Integrated Genome Browser (IGB).

### 4.8. Darkness-Induced Leaf Senescence Assay and Quantification of Chlorophyll Content

After seedlings were transplanted to soil, the 10-day-old plant rosette leaves were placed in a square petri dish containing three layers of filter paper, and the filter paper was immersed with deionized water, and the plate was wrapped tightly in aluminum foil, and placed in a growth chamber for cultivation. According to the method given in a previously published paper, we extracted chlorophyll from the leaf tissues of the plants using of 80% ice-cold acetone and measured and calculated chlorophyll content by UV spectrophotometer [88].

### 4.9. Published Data

The BS-seq data of NMR19-4 from all our used wild-type accessions were sourced from the 1001 Genomes (https://www.1001genomes.org, accessed on 10 April 2021). The data of ATAC-seq is available in the National Center of Biotechnology Information (NCBI) under the accession number SRP300093 (https://www.ncbi.nlm.nih.gov/sra/?term=SRP300093, accessed on 20 October 2021).

## 5. Conclusions

We have successfully used the CRISPR/dCas9-TET1cd system to edit NMR19-4 demethylation and obtained NMR19-4 demethylated plants. The demethylation status of NMR19-4 can be stably inherited in the progeny without the vector, and the CRISPR/dCas9-TET1cd system did not cause genome-wide methylation changes significantly; since NMR19-4 hypomethylation can upregulate the expression of *PPH* and promote leaf senescence, the results of quantitative experiments confirmed that the expression of *PPH* was up-regulated in NMR19-4 demethylated plants, and darkness-induced leaf senescence indicated that leaf senescence was accelerated in NMR19-4 demethylated plants. At the same time, the results of hybridization showed that demethylated NMR19-4 is Mendelian inheritance. These results provide a reference for our further spontaneous epialleles editing of crops in the future.

## Figures and Tables

**Figure 1 ijms-23-10492-f001:**
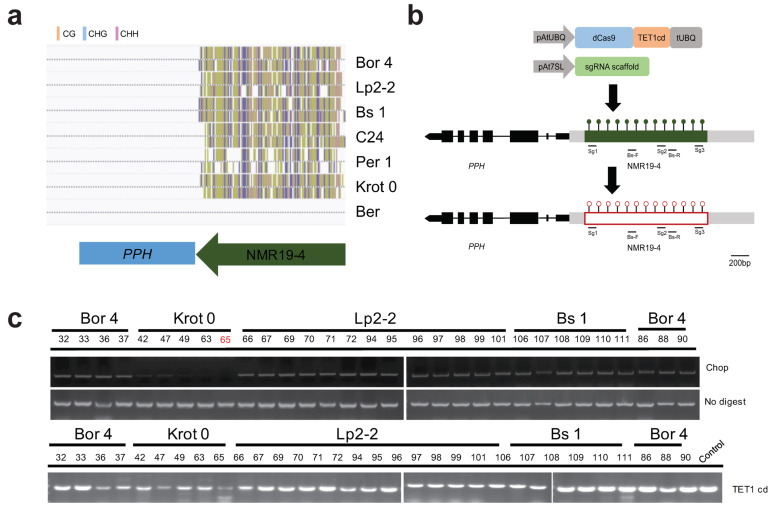
Targeted DNA demethylation of the Arabidopsis NMR19-4 using the CRISPR/dCas9-TET1cd epigenetic editing system. (**a**) Methylation level of NMR19-4 in Bor 4, Lp2-2, Bs 1, C24, Per 1, Krot 0, and Ber; (**b**) Schematic representation of the CRISPR/dCas9-TET1cd system. The target site and the regions for bisulfite PCR (BS-PCR) are shown in the schematic. Green filled circles indicate hypermethylation and red open circles indicate hypomethylation. (**c**) Chop-PCR analysis of the methylation level of NMR19-4 detected in transformation Arabidopsis T1 generation. The number marked in red represents the line we use in our subsequent experiments.

**Figure 2 ijms-23-10492-f002:**
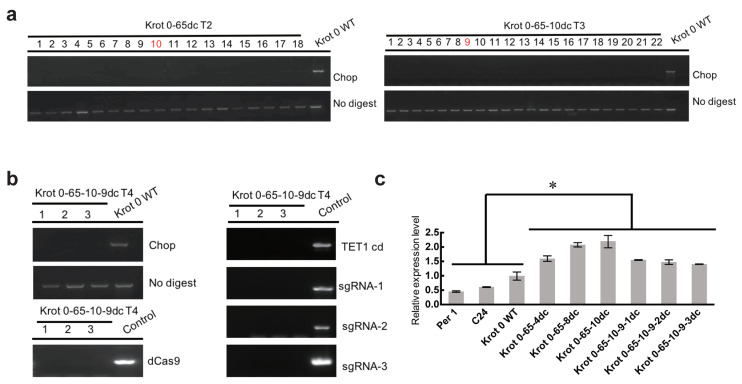
Multiple-generational inheritance of the epi-edited Krot 0-65dc. (**a**) Chop-PCR analysis of plants from T2 and T3 generations. The red numbers indicate the lines used in the experiment; (**b**) T4 generation without CRISPR/dCas9-TET1cd vector, the left is the results of T4 generation detection Chop-PCR and without dCas9, and the right is the T4 generation detection without TET1cd and sgRNAs. (**c**) *PPH* gene expression in Per 1, C24, Krot 0 WT (Krot 0 wild type), T2, and T4 generations (*n* = 3, *, *p* < 0.05, *t*-test).

**Figure 3 ijms-23-10492-f003:**
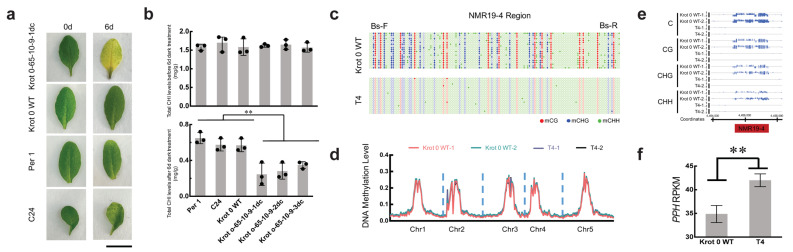
Darkness-induced leaf senescence and methylation status of NMR19-4 in epi-edited progenies. (**a**) Darkness-induced leaf senescence phenotypes in Krot 0-65-10-9-1dc, Krot 0 WT (Krot 0 wild-type), Per 1, and C24. Photos were taken from detached leaves before (0 day) and after 6 days darkness induction. (**b**) Quantification of chlorophyll content before (**top**) and after (**bottom**) 6 days darkness treatment in Per 1, C24, Krot 0 WT, Krot 0-65-10-9-1dc, Krot 0-65-10-9-2dc, and Krot 0-65-10-9-3dc (*n* = 3, Black dots represent different repetitions, **, *p* < 0.01, *t*-test). Chl represents chlorophyll. (**c**) Bisulfite sequencing analysis of Krot 0 WT (Krot 0 wild type) and T4 generation. The filled circles represent methylated cytosine and the empty circles represent unmethylated cytosine. (**d**) Genome methylation level between Krot 0 WT (Krot 0 wild type) and T4 generation. There are two replications in Krot 0 WT (Krot 0 wild type) and T4 generation. (**e**) Integrated Genome Browser (IGB) snapshot showing methylation variation between Krot 0 WT (Krot 0 wild type) and T4 generation generated from BS-seq data. The blue line indicates the level of methylation. (**f**) RPKM values from RNA-seq data of Krot 0 WT (Krot 0 wild type) and T4 generation (*n* = 3, **, *p* < 0.01, *t*-test).

**Figure 4 ijms-23-10492-f004:**
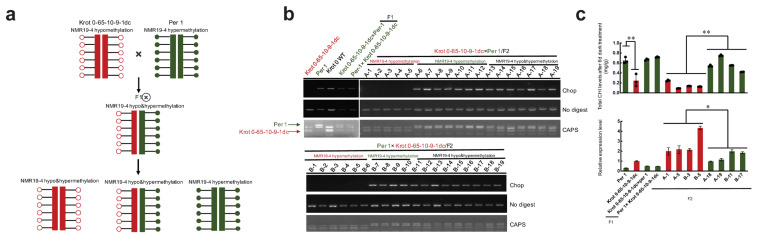
Demethylation-edited NMR19-4 loci can be trans-generationally transmitted by Mendelian inheritance in F1 hybrid and F2 segregation. (**a**) A schematic diagram of trans-generational transmission by Mendelian inheritance in F1 hybrid and F2 progeny. Filled green circles indicate hypermethylation and red empty circles indicate hypomethylation. (**b**) The methylation status of NMR19-4 was detected by Chop-PCR in F1 and F2 recombinant lines. The arrows on the left side of the gel photo indicate the specific alleles corresponding to the bands in the gel. The numbers of A1 to A19 and B1 to B19 represent the different individual plants, and all of them were used for Chop-PCR, CAPS, darkness-induced leaf senescence, and qPCR assays. CAPS: Based on the difference in the parental allele SNPs, the same primer was used for amplification, and the resulting PCR products were then digested with a restriction enzyme, and the specific inheritance allele in the progeny based on the distinguish of the digested band. (**c**) Quantification of chlorophyll content after 6 days of dark induction in the F1 and F2 recombinant lines by qPCR (*n* = 3) (**, *p* < 0.01, *t*-test) and the expression level of *PPH* in F1 and F2 recombinant lines by qPCR (*, *p* < 0.05, *t*-test).

## Data Availability

Not applicable.

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
