# Peer review of "Transgenerationally Transmitted DNA Demethylation of a Spontaneous Epialleles Using CRISPR/dCas9-TET1cd Targeted Epigenetic Editing in Arabidopsis"

_ijms, 2022, doi:10.3390/ijms231810492_

Round 1
Reviewer 1 Report
In a previous study, Zhang group had identified a spontaneous naturally occurring methylation region (NMR19-4) located within the promoter region of the PPH (Pheophytin Pheophorbide Hydrolase) gene in Arabidopsis. To further demonstrate the methylation status of NMR19-4 is highly associated with the expression level of PPH, the authors used a previously reported CRISPR/dCas9-TET1cd system to target the hypermethylated NMR19-4 region using three sgRNAs. They found that CRISPR/dCas9-TET1cd could specific reduce the methylation level at the target sites, which further enhanced the PPH expression and accelerated leaf senescence. In addition, hybridization assays confirmed that the methylation status of NMR19-4 adhered the Mendelian inheritance, suggesting spontaneous epialleles might be transgenerationally transmitted.
Overall, this work demonstrated the correlation between the methylation status of NMR19-4 and the expression level of PPH, providing new insights into epigenetic mechanisms underlying plant development. More importantly, this work also shed light on the transgenerational inheritance of spontaneous epialleles. I would recommend publishing this study in the International Journal of Molecular Sciences after the authors address my concerns.
Major concerns:
Lines 168 to 170:
It would be better to have a control vector dCas9 without the TET1cd for ecotype Krot 0 transformation. This control could exclude the possibility of the cutting differences between Krot 0 and other ecotypes are not caused by dCas9 binding. More importantly, this control can further confirm that the change of PPH mRNA truly results from the TET1cd-mediated demethylation, especially in the T2 and T3 generations.
Lines 210 to 212:
Was the dark-induced senescence identified in T2 and T3 plants? Figure 2c showed the expression level of PPH in T2 generation is higher than that of T4 plants. Based on the authors’ assumption, T2 plants should have a stronger phenotype.
Minor concerns:
Lines 156 to 158: the authors selected four NMR19-4 hypermethylated ecotypes
for demethylation editing. However, Figure 1a showed six ecotypes besides the control ecotype Ber. It is kindly confused.
Line 77: “small guide RNA (sgRNA)”. Single guide RNA (sgRNA) is more accurate.
Remove the unnecessary dash within one word. E.g. lines 132 “hyper-methylated”, 134 “with-out”... Please go through the manuscript and correct all of them.
Reviewer 2 Report
Very interesting work Transgenerationally Transmitted DNA Demethylation of a Spontaneous Epialleles using CRISPR/dCas9-TET1cd Targeted Epigenetic Editing in Arabidopsis. I suggest that the authors add a perspective section on the basis of these obtained results.
Reviewer 3 Report
The authors presented a good approach in the manuscript entitled “Transgenerationally Transmitted DNA Demethylation of a Spontaneous Epialleles using CRISPR/dCas9-TET1cd Targeted Epigenetic Editing in Arabidopsis”. The study topic is interesting and informative. However, the main concerns about this manuscript can be found below, and major revision is suggested.
Authors should improve the overall English of the manuscript. Please check spelling, grammar, spacing, and punctuation properly. The language is not satisfactory to be published at this stage in this journal.
1) It could better reflect the title if you use the complete generic name of the plant in the title
2) Please check the authors’ address and correct the font style (especially line number 14) and set it properly.
3) Read journal submission guidelines properly and mention the figure with capital letters throughout the MS.
4) In line number 264, give between RdDM and citation number.
5) In line number 301, remove the space between citation number according to Journal submission guidelines.
6) In line number 402, is it CRISPR/Cas9-TET1 or CRISPR/Cas9-TET1cd? Clarify
7) In line number 437 and 464, Enclose Tables in Brackets.
8) In line number 477, italicize the scientific name
9) Please check the References are correct, italicize the scientific names, and correct the dashes.
10) Dash (-) in the volume of all references does not match with journal requirements. Kindly correct it.
TRANSLATE with x English
| Arabic | Hebrew | Polish |
| Bulgarian | Hindi | Portuguese |
| Catalan | Hmong Daw | Romanian |
| Chinese Simplified | Hungarian | Russian |
| Chinese Traditional | Indonesian | Slovak |
| Czech | Italian | Slovenian |
| Danish | Japanese | Spanish |
| Dutch | Klingon | Swedish |
| English | Korean | Thai |
| Estonian | Latvian | Turkish |
| Finnish | Lithuanian | Ukrainian |
| French | Malay | Urdu |
| German | Maltese | Vietnamese |
| Greek | Norwegian | Welsh |
| Haitian Creole | Persian |
TRANSLATE with EMBED THE SNIPPET BELOW IN YOUR SITE Enable collaborative features and customize widget: Bing Webmaster Portal Back
Reviewer 4 Report
In this study, Wang et.al specifically demethylated a spontaneous epiallele of NMR19-4 in the hypermethylated ectotype Krot 0 by the CRISPR/dCas9-TET1cd system. The demethylated plants showed higher expression of PPH and accelerated leaf senescence, confirming the critical role of NMR19-4 methylation status in determining PPH expression. Particularly, the authors found that the artificial demethylated NMR19-4 is stable through generations and showed Mendelian inheritance. In general, I think this is an interesting study and a useful attempt for plant genome epi-editing.
There are a number of suggestions for this manuscript, which are listed below.
1. In Figure 2c, the expression of PPH in T4 generation plants seems lower than T2 plants. Is it true that the demethylated plants eventually recover the methylation status to its native levels by multiple generations?
2. Whether the methylation status of NMR19-4 changes in the processes of age- and/or dark-induced leaf senescence. If so, will the change be inherited by offspring plants?
3. In the Mendelian inheritance assays, the hypermethylation phenotype seems dominant in the F1 and F2 plants, please explain why.
4. The demethylation status can be stably maintained by transgenerational inheritance when crossed with different genotypes. What about the situation when Krot 0-65dc cross with the wild-type Krot 0?
5. PCR results of non-transgenic hypermethylated wild-type ecotypes and control hypomethylated ecotype Ber should be included in Figure 1c.
Minor points:
1. Page 2, line 60, the full name of DRM2 has been mentioned in the above (Page 2, line 53);
2. Page 3, line 150, the full name of “PPH” should be given at the first time it is mentioned;
3. Page 4, line 152, delete “and conversely”;
4. Page 4, line 161, “Pol iii” should be “Pol III”;
5. The captions of y-axis is too small in Figure 3b and 4c;
6. Page 7, line 273, “Figure S2” should be “Figure S3”;
7. Page 7, line 293, delete one “our previous”;
8. Page 10, line 421, “seedinds” should be “seedlings”;
9. The reference No.52 is not accurate;
10. There are some grammatical errors and typos throughout the manuscript. Please do a careful copyediting before resubmission of a revised manuscript.
Round 2
Reviewer 1 Report
The authors have addressed most of my concerns. This revised manuscript is ready for publishing.
Reviewer 3 Report
Thanks to all the authors for incorporating all the suggested changes. I believe the manuscript has been sufficiently improved for publication in IJMS.
Reviewer 4 Report
The authors have addressed most of my queries, I recommend it published on IJMS. But for my Point 3 “In the Mendelian inheritance assays, the hypermethylation phenotype seems dominant in the F1 and F2 plants, please explain why.” I don’t think the authors have given sufficient explanations. The cartoon exhibited by authors is not appropriate for my question, as in the F1 hybrid, the hypermethylated and hypomethylated DNA fragments of NMR19-4 are separated on homologous chromosomes. Will the hypermethylated locus of NMR19-4 affects the transcription of the hypomethylated locus of NMR19-4 on another chromosome? I guess the transcription of homologous loci on different chromosomes are independently regulated by their methylation status. In this scenario, the hypomethylated loci of NMR19-4 in the F1 hybrid can give high expression of PPH, however, this expression level may not sufficient for proper PPH activities.